# Impaired motor-to-sensory transformation mediates auditory hallucinations

**Fuyin Yang**[1,2,3], **Hao Zhu**[2,4], **Xinyi Cao**[1], **Hui Li**[1], **Xinyu Fang**[1], **Lingfang Yu**[1], **Siqi Li**[2,3], **Zenan Wu**[1], **Chunbo Li**[1,5], **Chen Zhang**[1]*, **Xing Tian**[2,3,4]*

1 Shanghai Key Laboratory of Psychotic Disorders, Shanghai Mental Health Center, Shanghai Jiao Tong University School of Medicine, Shanghai, China, 2 NYU-ECNU Institute of Brain and Cognitive Science at NYU Shanghai, Shanghai, China, 3 Shanghai Key Laboratory of Brain Functional Genomics (Ministry of Education), School of Psychology and Cognitive Science, East China Normal University, Shanghai, China, 4 Shanghai Frontiers Science Center of Artificial Intelligence and Deep Learning; Division of Arts and Sciences, New York University Shanghai, Shanghai, China, 5 Institute of Psychology and Behavioral Science, Shanghai Jiao Tong University, Shanghai, China

* zhangchen645@gmail.com (CZ); xing.tian@nyu.edu (XT)

**Data Availability Statement:** All EEG dataset, analysis codes and stimulation files are available from the OSF database (http://osf.io/rsnu4/).

## Abstract

Distinguishing reality from hallucinations requires efficient monitoring of agency. It has been hypothesized that a copy of motor signals, termed *efference copy (EC)* or *corollary discharge (CD)*, suppresses sensory responses to yield a sense of agency; impairment of the inhibitory function leads to hallucinations. However, how can the sole absence of inhibition yield positive symptoms of hallucinations? We hypothesize that selective impairments in functionally distinct signals of *CD* and *EC* during motor-to-sensory transformation cause the positive symptoms of hallucinations. In an electroencephalography (EEG) experiment with a delayed articulation paradigm in schizophrenic patients with (AVHs) and without auditory verbal hallucinations (non-AVHs), we found that preparing to speak without knowing the contents (general preparation) did not suppress auditory responses in both patient groups, suggesting the absent of inhibitory function of *CD*. Whereas, preparing to speak a syllable (specific preparation) enhanced the auditory responses to the prepared syllable in non-AVHs, whereas AVHs showed enhancement in responses to unprepared syllables, opposite to the observations in the normal population, suggesting that the enhancement function of *EC* is not precise in AVHs. A computational model with a virtual lesion of an inhibitory inter-neuron and disproportional sensitization of auditory cortices fitted the empirical data and further quantified the distinct impairments in motor-to-sensory transformation in AVHs. These results suggest that "broken" *CD* plus "noisy" *EC* causes erroneous monitoring of the imprecise generation of internal auditory representation and yields auditory hallucinations. Specific impairments in functional granularity of motor-to-sensory transformation mediate positivity symptoms of agency abnormality in mental disorders.

## Introduction

Perceptual experiences can be induced by sensory processing [1,2] as well as constructed without external stimuli, such as memory retrieval [3] and mental imagery [4–8]. Such distinct

**Funding:** This study was supported by the National Natural Science Foundation of China 32071099 and 32271101 (https://www.nsfc.gov.cn/), Natural Science Foundation of Shanghai 20ZR1472100 (https://svc.stcsm.sh.gov.cn/), Program of Introducing Talents of Discipline to Universities, Base B16018 to X.T., East China Normal University (ECNU) Academic Innovation Promotion Program for Excellent Doctoral Students YBNLTS2019-026 (http://www.yjsy.ecnu.edu.cn/ ) and China Postdoctoral Foundation under Grant Number 2024M752047 (https://www.chinapostdoctor.org. cn/bshjjh/) to F.Y. The funders had no role in study design, data collection and analysis, decision to publish, or preparation of the manuscript.

**Competing interests:** The authors have declared that no competing interests exist.

**Abbreviations:** AHRS, Auditory Hallucinations Rating Scale; AVH, auditory verbal hallucination; CD, corollary discharge; DDD, defined daily dose; EC, efference copy; EEG, electroencephalography; ERP, event-related potential; GFP, global field power; GP, general preparation; LSD, least significant difference; PANSS, Positive and Negative Syndrome Scale; RT, reaction time; SP, specific preparation.

causes of perceptual experiences necessitate efficient monitoring to distinguish the inducing sources; failure of monitoring may result in hallucinations [9,10]. For example, patients with auditory hallucinations, the core symptoms of schizophrenia, often "hear" voices in the absence of sound [11]. Patients may fail to distinguish between their thoughts (e.g., inner speech, [12]) and external voices, resulting in a reduced ability to recognize thoughts as self-generated [13,14]. That is, the symptoms of hallucinations have been attributed to the malfunction of self-monitoring [15,16].

Self-monitoring can be achieved with the mechanism of internal forward models [17,18], in which a copy of motor signals, termed corollary discharge (CD) [19] or efference copy (EC) [20], transmits to sensory regions (motor-to-sensory transformation) and suppresses sensory neural activities. The internal forward models have been evident ubiquitously across the animal kingdom [21] and the sensory suppression has been hypothesized as an index for signaling the impending reafference as the consequence of an agent's actions—the sense of agency [22–25]. That is, the inhibitory function of motor signals may provide an automatic computation to distinguish the sensory neural responses that are either internally induced or evoked by external stimulation [22,26]. Impairment of the inhibitory functions in motor-to-sensory transformation may result in self-monitoring malfunction and lead to auditory hallucinations [16,27,28].

However, how can the monitoring function of the agency inhibit sensory processing while at the same time constructing the positive symptoms of auditory hallucinations? Hallucinations are perceptual-like experiences that require specific neural representation activated without sensory stimulation [29]. The sole inhibitory function of the motor copies cannot fully explain the symptoms of auditory hallucinations and is challenged by recent empirical findings. For example, action-induced enhancement has been found in a subset of sensory cortices in addition to action-induced suppression [30–34]. Auditory neural representations are constructed in auditory working memory based on covert speaking [8,35,36]. This motor-based auditory working memory (i.e., inner speech) that activates specific neural representation may be misattributed to external sources because of the impaired self-monitoring function in schizophrenia [37,38]. All these recent results hint that the copy of motor signals may have a function that sensitizes the sensory cortices in addition to the inhibitory function for monitoring agency. The combination of 2 complementary functions may mediate the positive symptoms of auditory hallucinations.

A recent study provided preliminary evidence supporting a hypothesis of distinct modulatory functions of motor signals on perceptual processes [39]—the CD function is generic motor discharge available throughout the entire time course of action and can inhibit processes in the connected sensory regions for indicating all possible sensory consequences of actions [40] (Fig 1A, cyan arrow). Whereas the EC function is a copy of a specific motor plan and selectively enhances the sensitivity to sensory reafference targets caused by actions (Fig 1A, red arrow). Based on this theoretical framework, in this study, we hypothesize that selective impairments in CD and EC functions mediate the positive symptoms of auditory hallucinations. Specifically, the CD inhibitory function that should be available in the early stage of motor intention does not operate normally in all schizophrenia patients (Fig 1B and 1C, gray arrows)—the negative symptoms (e.g., lack of desire) in patients without auditory verbal hallucinations (non-AVHs) may have weaker movement intention and hence diminish CD signals at the beginning stage of speaking (Fig 1B, only left part of CD arrow is gray); whereas in patients with AVHs, in addition to the negative symptoms, the deficit in the self-monitoring of agency is manifested in the impairment of the inhibitory function of CD throughout the entire course of action—a "broken" CD (Fig 1C, the entire CD arrow is gray). When EC is available after specific movement plans have been formed, the enhancement function of EC on

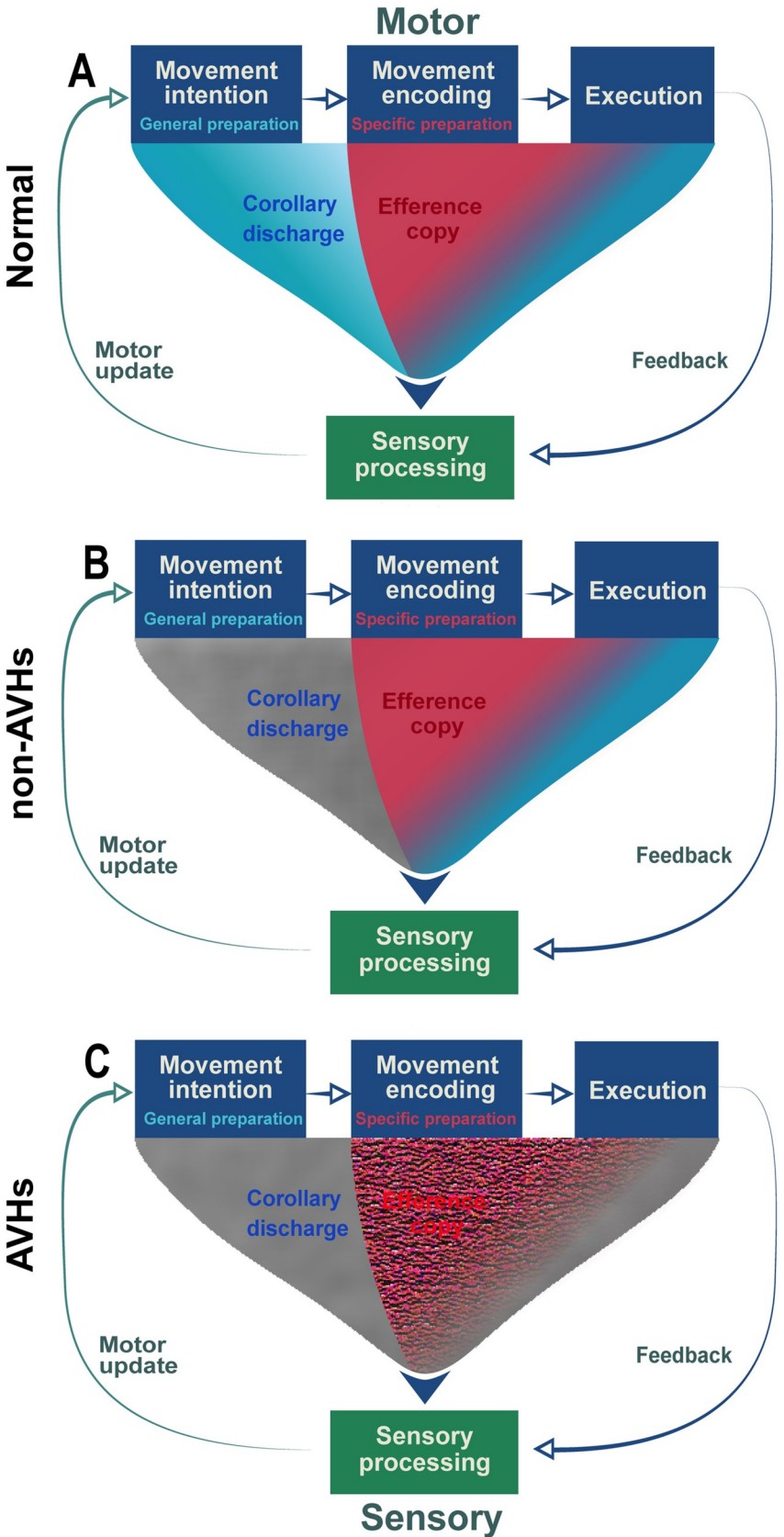

**Fig 1. Schematics of distinct functions of motor signals in motor-to-sensory transformation across temporal stages of action in normal and clinical populations.** (A) The distinct inhibition and enhancement functions in motor-to-sensory transformation in the normal population. CD is a generic discharge signal from the motor system that does not necessarily include any content information. The CD is available at all stages of motor processes and can onset as early as in motor intention (i.e., general preparation, for example, preparing to speak without knowing what to say). The function of CD is inhibiting all sensory regions that are connected with the activated motor system, indicating the impending sensory consequences of actions and hence yielding the sense of agency (Fig 1A, cyan arrow). EC, a duplicate of the planned motor signals, is available during motor encoding (i.e., specific preparation, for example, preparing to speak a specific speech). The copy of detailed action codes selectively boosts the sensitivity of neural responses to the sensory target of actions (Fig 1A, red arrow). (B) The intact enhancement but possible deficits in inhibition functions in motor-to-sensory transformation in schizophrenia patients without auditory verbal hallucinations (non-AVHs). The negative symptoms of non-AVHs patients may cause weak motor intention that leads to diminished inhibitory function of CD at the beginning stage of an action (Fig 1B, gray in the left part of the CD arrow), whereas the function of CD in the following motor processes (Fig 1B, blue in the right part of the CD arrow, identical to that in Fig 1A) as well as the specific modulation effects of enhancement in EC are preserved (Fig 1B, red arrow, identical to that in Fig 1A). (C). The impaired inhibitory function of CD and imprecise enhancement function of EC in schizophrenia patients with AVHs. The malfunctioned monitoring of agency in AVHs is mediated by the impaired inhibitory function of CD throughout the entire time course of motor processes (Fig 1C, entire gray arrow). Moreover, the positive symptoms of AVHs—random perceptual-like experiences without corresponding acoustic stimulations—would be mediated by imprecise EC (Fig 1C, hatched red arrow) that could activate multiple auditory neural representations around the sensory target of the action. That is, the positive symptoms of AVHs are an emergent property of the impaired sensorimotor systems in which a "broken" CD misattributes the inducing sources of the auditory neural representations that are activated by a "noisy" EC without external stimulations. Fig 1A reproduced with permission from the original publisher [39] and Oxford University Press (license number 5858081359804). AVH, auditory verbal hallucination; CD, corollary discharge; EC, efference copy.

the prepared speech target is intact in non-AVHs patients (Fig 1B, red arrow). Whereas, in patients with AVHs, the EC enhancement function is imprecise (Fig 1C, hatched red arrow) and modulates both target reafference and its neighboring auditory units—a "noisy" EC (potential causes of various perceptual-like auditory and verbal contents during hallucinations). That is, the positive symptoms of auditory hallucinations are mediated by the combination of a "broken" CD and "noisy" EC. This hypothesis of distinct impairments in motor-to-sensory transformation predicts that (1) the suppression effects of general speech preparation (e.g., preparing to speak without knowing what to say) that was observed in normal population would be absent in both patients with and without AVHs; and (2) non-AVHs patients would show identical enhancement effects of specific speech preparation (e.g., preparing to speak a given syllable) as in normal population, but the modulation effects of specific speech preparation would be different in patients with AVHs from those in non-AVHs and normal population.

## Result

### Demographic and clinical data

S1 Table shows the participants' demographic data and the clinical variables. One-way ANOVA analyses revealed a significant difference among 3 groups (AVHs, non-AVHs, and normal) in the GP task for age ($F_{(2,56)} = 8.06$, $p = 0.001$), education ($F_{(2,56)} = 8.87$, $p < 0.001$) and in the SP task for age ($F_{(2,53)} = 8.40$, $p = 0.001$), education ($F_{(2,53)} = 4.64$, $p = 0.014$). Fisher's LSD post hoc tests revealed no significant differences between AVHs and non-AVHs groups in education in GP and SP tasks (all $p > 0.05$). There was no significant difference in height and weight among the 3 groups (all $p > 0.05$). The chi-square test showed no significant differences among the 3 groups in gender. Further, the positive symptom scores ($t_{(1,38)} = 2.75$, $p = 0.009$) and P3 subscore ($t_{(1,38)} = 11.82$, $p < 0.001$) were significantly higher in the AVHs group than in the non-AVHs group. The negative symptom scores and general psychopathology scores as well as Positive and Negative Syndrome Scale (PANSS) total scores

were not significantly different in AVHs and non-AVHs groups. Neither the age of onset nor the duration was significantly different between the 2 groups.

## Behavioral preparation effects in the delayed articulation task

In the AVHs group (Fig 2C), a repeated-measure one-way ANOVA on RTs found a significant main effect of preparation ($F(3,57) = 80.04$, $p < 0.01$, partial $\eta^2 = 0.51$). Further analysis revealed that the onset of articulation was consistently faster after preparation. Specifically, RTs of articulation were the fastest after SP (mean = 450.34 ms; SD = 22.03) among all 4 conditions ($F(3,76) = 26.17$, $p < 0.01$, partial $\eta^2 = 0.51$). Articulation after GP (mean RT = 634.24 ms; SD = 17.89) was faster than immediate vocalization without preparation (NP, mean RT = 748.88 ms; SD = 33.71) ($t(19) = 5.43$, $p < 0.01$, $d = 0.93$). RTs were also significantly shorter when participants performed GP without sound probes (GP$_{NS}$, mean RT = 649.29 ms; SD = 17.52) than NP ($t(19) = 4.47$, $p < 0.01$, $d = 0.81$). These behavioral results suggested that participants engaged in speech preparation. More importantly, RTs were not significant between GP and GP$_{NS}$ ($t(19) = 1.83$, $p = 0.08$, $d = 0.19$). These results suggested that participants prepared the articulation based on the visual cues and the corollary discharge was available in the general preparation (GP) stage.

In the non-AVHs group (Fig 2D), the statistical results were similar to the ones in the AVHs group. A repeated-measure one-way ANOVA on RTs found a significant main effect of preparation ($F(3,57) = 128.99$, $p < 0.01$, partial $\eta^2 = 0.59$). Further analysis revealed that the onset of articulation was consistently faster after preparation. Specifically, RTs in SP (mean RT = 414.94 ms; SD = 21.57) were fastest among all 4 conditions ($F(3,76) = 37.03$, $p < 0.01$, partial $\eta^2 = 0.59$). RTs in GP (mean RT = 609.84 ms; SD = 19.96) was faster than in NP (mean RT = 708.40 ms; SD = 19.32) ($t(19) = 5.92$, $p < 0.01$, $d = 1.09$). RTs were also significantly shorter in GP$_{NS}$ (mean RT = 623.10 ms; SD = 18.49) than in NP ($t(19) = 5.58$, $p < 0.01$, $d = 0.98$). These results suggested that participants engaged in speech preparation. Moreover, the RTs were not significantly different between GP and GP$_{NS}$ ($t(19) = 1.99$, $p = 0.06$, $d = 0.15$). These results suggested that participants performed the GP task according to the visual cues and *CD* was available during the GP stage. These consistent behavioral results confirmed that both groups of patients can perform the behavioral preparation tasks.

## The impaired function of motor signals during general preparation

We first performed within-subject statistical analyses of paired *t* tests on the ERPs to the auditory probes in GP to investigate the modulatory effects of CD signals on auditory processes in each patient group. In the AVHs group, the N1 and P2 topographies showed typical auditory response patterns in both GP and B conditions (Fig 3A). However, the magnitude of neural responses in GP was not significantly different from B, neither in the early auditory responses of N1 component ($t(19) = 2.39$, $p = 0.054$, $d = 0.47$) nor in the later auditory responses of P2 component ($t(19) = 0.97$, $p = 0.43$, $d = 0.10$) (Fig 3B). These results contrasted with the ones obtained in the normal population in which GP suppressed auditory responses (S1 Fig) [39]. These results supported the hypothesis that the function of CD was impaired in AVHs patients.

Because of the costs and efforts to conduct a direct replication of the predicted and observed null results in the GP condition, we implemented a bootstrapping procedure to estimate the reliability of the null results. The simulation results (S2 Fig) showed that the suppression (modulation index value less than 0) was outside the 99% confidence interval, suggesting that the observed null results in the AVH group were highly unlikely due to chance.

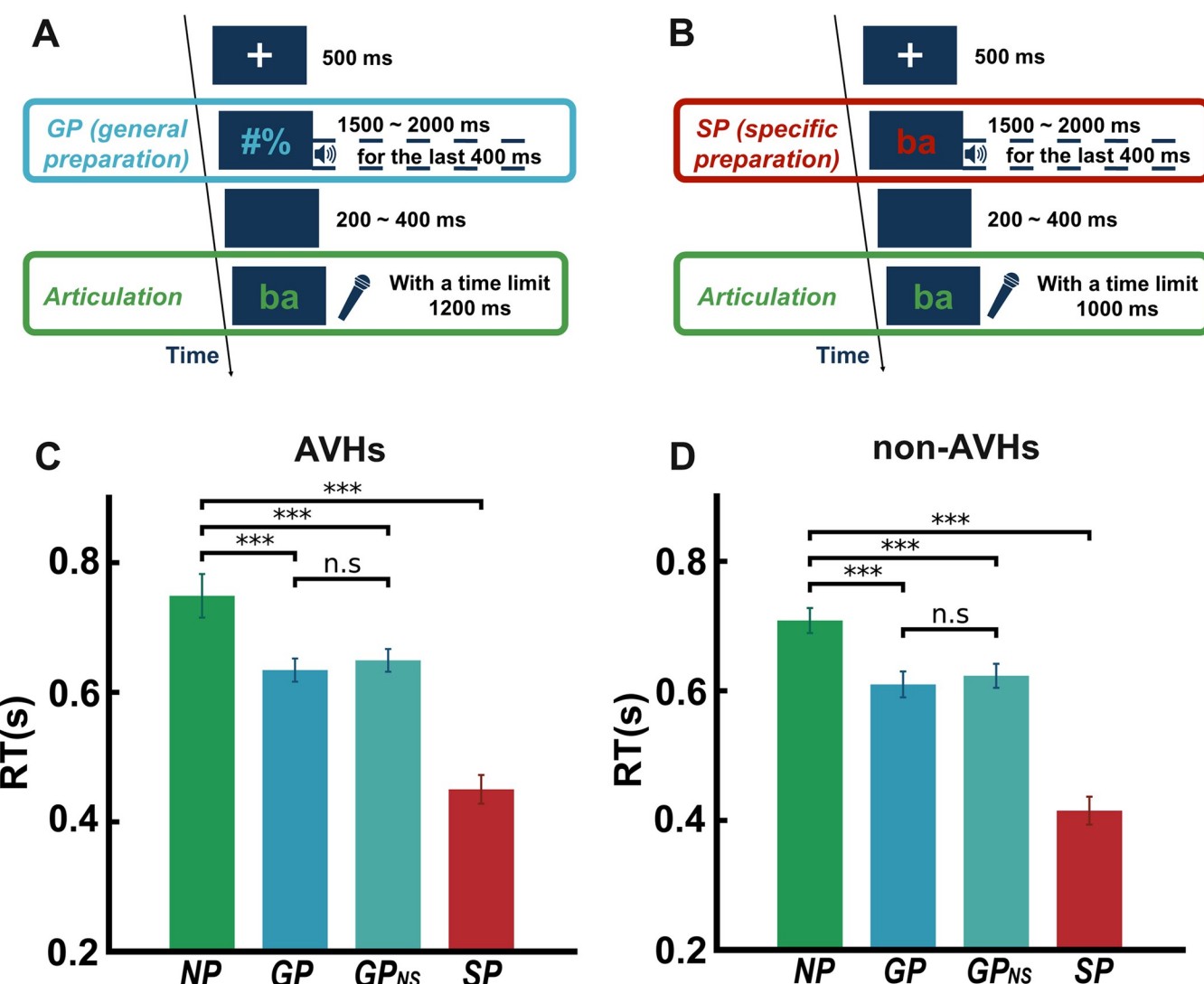

**Fig 2. Experimental paradigm and behavioral results in AVHs and non-AVHs patient groups.** (A) Illustration of a sample trial of GP. After a fixation displayed for 500 ms, a yellow visual cue of 2 symbols (#%) appeared in the center of the screen for a duration in a range between 1,500 ms to 2,000 ms with an increment of 100 ms. Participants were asked to prepare to speak in the upcoming articulation task, although they did not know what to say because the symbols did not contain any linguistic information. In half of the trials, an auditory probe, either one of the 4 auditory syllables (/ba/, /pa/, /ga/, and /ka/) or a 1 kHz pure tone, was presented during the last 400 ms of the preparatory stage. Another half of the trials did not include any auditory probes (GP$_{NS}$). After a blank screen with a range of 200 ms to 400 ms with an increment of 50 ms, participants saw a green visual cue that was one of the 4 syllables (/ba/, /pa/, /ga/, and /ka/) in the center of the screen for a maximum of 1,200 ms and were asked to produce the syllable as fast and accurately as possible. (The experimental scripts and auditory stimuli can be found at http://osf.io/rsnu4/.) (B) Illustration of a sample trial of SP. The procedure was similar to the GP task with 2 exceptions: (1) the visual cue during the preparatory stage was a red syllable randomly selected from the 4 syllables (/ba/, /pa/, /ga/, and /ka/), and (2) an auditory probe was presented in every trial during the preparatory stage. The auditory probes were either the same as or different from the visual cue, yielding 2 conditions—auditory syllables were congruent (SPcon) or incongruent (SPinc) with the syllable that participants prepared to speak. (C, D) Behavioral results of AVHs and non-AVHs patients. The speed of articulation was measured as RT. In both groups, the RTs in GP (with or without auditory probes) and SP were significantly faster than those in NP, suggesting that preparation was carried out in both groups and all conditions. Error bars indicate ± SEM. ∗∗$P < 0.01$, ∗∗∗$P < 0.001$. (The underlying RT data for this figure can be found at http://osf.io/rsnu4/ and in S1 Data.) AVH, auditory verbal hallucination; GP, general preparation; RT, reaction time; SP, specific preparation.

In the non-AVHs group, the results were similar to those in the AVHs group. The N1 and P2 topographies showed typical auditory response patterns in both conditions (Fig 3C). The magnitude of neural responses in GP was not significantly different from B, neither in N1 ($t$ (19) = 0.12, $p = 0.91$, $d = 0.02$) nor in P2 ($t$(19) = 1.96, $p = 0.09$, $d = 0.34$) (Fig 3D). These results

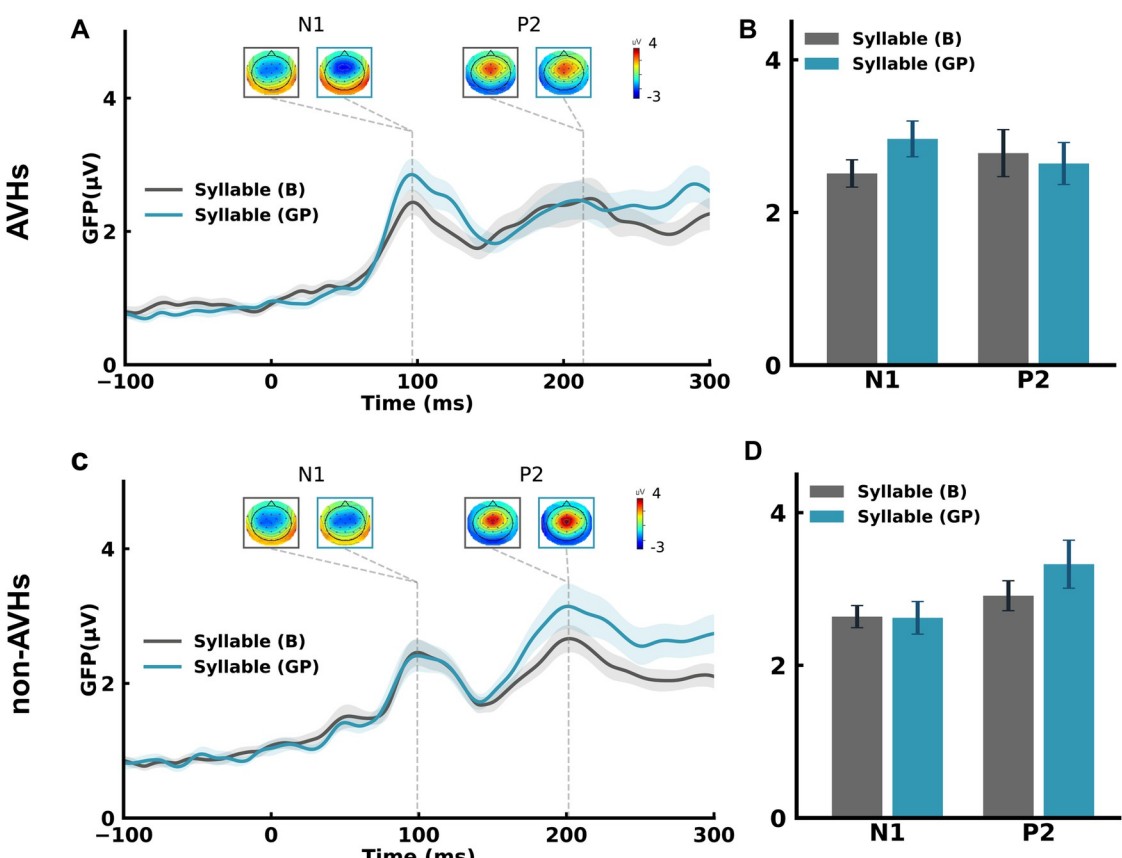

**Fig 3. The absence of modulation effects on auditory responses during GP in both AVHs and non-AVHs groups.** (A) ERP time course and topographic responses for GP and B conditions in AVHs patients. Peak amplitudes and latencies of the N1 and P2 components were observed in the GFP waveform for each condition. The response topographies at each peak latency are shown in boxes with the same color code for each condition. (B) Mean GFP amplitudes at N1 and P2 latencies in GP (blue) and B (gray) conditions in AVHs patients. (C) ERP time course and topographic responses in GP and B conditions in non-AVHs patients. (D) Mean GFP amplitude at the N1 and P2 latencies in GP (blue) and B (gray) conditions in non-AVHs patients. No significant differences between GP and B were observed in either group. Error bars indicate ± SEMs. (The underlying ERP data for this figure can be found at http://osf.io/rsnu4/ and in S2 Data.) AVH, auditory verbal hallucination; ERP, event-related potential; GP, general preparation.

suggest that the effects of CD during the earliest stage of motor intention were also absent in non-AVHs patients.

We further performed two-way mixed ANOVAs to assess the differences in N1 among groups. The group (AVHs, non-AVHs, and normal) was a between-subject factor, and the condition (B and GP) was a within-subject factor. The main effect of the group was not significant ($F_{(2,56)} = 1.91$, $p = 0.157$, partial $\eta^2 = 0.06$), neither was the main effect of condition ($F_{(1,56)} = 0.63$, $p = 0.43$, partial $\eta^2 = 0.01$). Whereas the interaction of group and condition was significant ($F_{(2,56)} = 6.28$, $p = 0.003$, partial $\eta^2 = 0.18$). The significant interaction was further explored by separate ANOVA for each pairwise group comparison (AVHs versus normal, non-AVHs versus normal, and AVHs versus non-AVHs). The group and condition interaction was significant for the comparison of AVHs and normal ($F_{(1,37)} = 11.65$, $p = 0.002$, partial $\eta^2 = 0.24$). However, no significant interaction was observed for the comparison between AVHs and non-AVHs ($F_{(1,38)} = 4.02$, $p = 0.052$, partial $\eta^2 = 0.1$), and non-AVHs and normal ($F_{(1,37)} = 2.28$, $p = 0.14$, partial $\eta^2 = 0.06$). These results statistically supported the absence of

CD inhibitory effects during the earliest stage of action (i.e., motor intention, manifested by general preparation) in both AVHs and non-AVHs groups.

To explore the modulation functions of CD signals on tones, we conducted paired $t$ tests on the auditory responses to tones between GP and B for N1 and P2 separately. In the AVHs group, the effects were not significant neither in N1 ($t(19) = 0.56$, $p = 0.94$, $d = 0.06$) nor in P2 ($t(19) = 0.32$, $p = 0.76$, $d = 0.07$) (S3A and S3B Fig). In the non-AVHs group, the effects in N1 ($t(19) = 1.04$, $p = 0.47$, $d = 0.20$) and P2 ($t(19) = 0.88$, $p = 0.59$, $d = 0.18$) were not significant (S3C and S3D Fig). The N1 and P2 topographies showed typical auditory response patterns in both groups and conditions (S3A and S3C Fig). The absence of modulation effects on tones in AVHs and non-AVHs patients contrasted with the increased error responses of N1 in normal controls [39], further supporting the hypothesis of impaired CD during motor intention in schizophrenia patients.

## The functions of motor signals during specific preparation dissociated between AVHs and non-AVHs

We next investigated the function of EC on the ERPs to the auditory probes in SP. In the AVHs group, the N1 and P2 topographies showed typical auditory response patterns among SPinc, SPcon, and B conditions (Fig 4A). The magnitude of N1 was larger than that in B when the auditory syllables were incongruent with the contents of preparation in SP (SPinc) ($t(19) = 3.69$, $p = 0.004$, $d = 0.72$). The effect was not significant in the later auditory responses of P2 ($t(19) = 1.69$, $p = 0.43$, $d = 0.22$). However, when the auditory syllables were congruent with the specific preparation (SPcon), the effect was not significant in N1 ($t(19) = 1.26$, $p = 0.30$, $d = 0.23$) nor in P2 ($t(19) = 1.52$, $p = 0.43$, $d = 0.20$) (Fig 4B). These results were opposite to the results in normal controls in which motor signals during SP enhanced the perceptual responses to the congruent auditory syllable probes (S4 Fig) [39]. The observed modulation effects in AVHs suggested that EC was available during specific preparation, but the opposite modulation patterns (enhancement in SPinc in AVHs compared with enhancement in SPcon in normal) indicated a "noisy" EC that yielded imprecise modulation on auditory responses during specific preparation in AVHs.

In the non-AVHs group, the N1 and P2 topographies showed typical auditory response patterns in all conditions (Fig 4C). When the auditory syllables were congruent with the specific preparation (SPcon), the response magnitude of the N1 component was larger than that in B ($t(19) = 3.18$, $p = 0.01$, $d = 0.56$), whereas the response magnitude of P2 component was reduced relative to B ($t(19) = 3.07$, $p = 0.02$, $d = 0.62$). In SPinc, the response magnitude of N1 ($t(19) = 0.73$, $p = 0.64$, $d = 0.13$) and P2 ($t(19) = 1.91$, $p = 0.09$, $d = 0.44$) was not significantly different from B (Fig 4D). The results of SP in non-AVHs were consistent with the results from the normal control group (S4 Fig) [39], suggesting an intact EC function in non-AVHs.

To further investigate the different modulation patterns of EC in SPcon and SPinc conditions among 3 groups, we calculated a difference score by subtracting the N1 response amplitude in the SPinc condition from the SPcon condition in 3 groups (AVHs, non-AVHs, and normal). The difference scores were subject to a one-way ANOVA with the group as a between-subject factor. The results revealed a different effect among the 3 groups ($F(2,53) = 21.52$, $p < 0.01$, partial $\eta^2 = 0.45$). Further post hoc $t$ tests showed that the different scores of N1 responses were significantly different between AVHs and non-AVHs ($t(15) = 5.33$, $p < 0.01$, $d = 2.29$), as well as between AVHs and normal ($t(15) = 4.15$, $p = 0.001$, $d = 1.30$), and between non-AVHs and normal ($t(15) = 2.53$, $p = 0.023$, $d = 0.91$). These results further suggested that the modulating effects of EC were distinct between AVHs and non-AVHs patients.

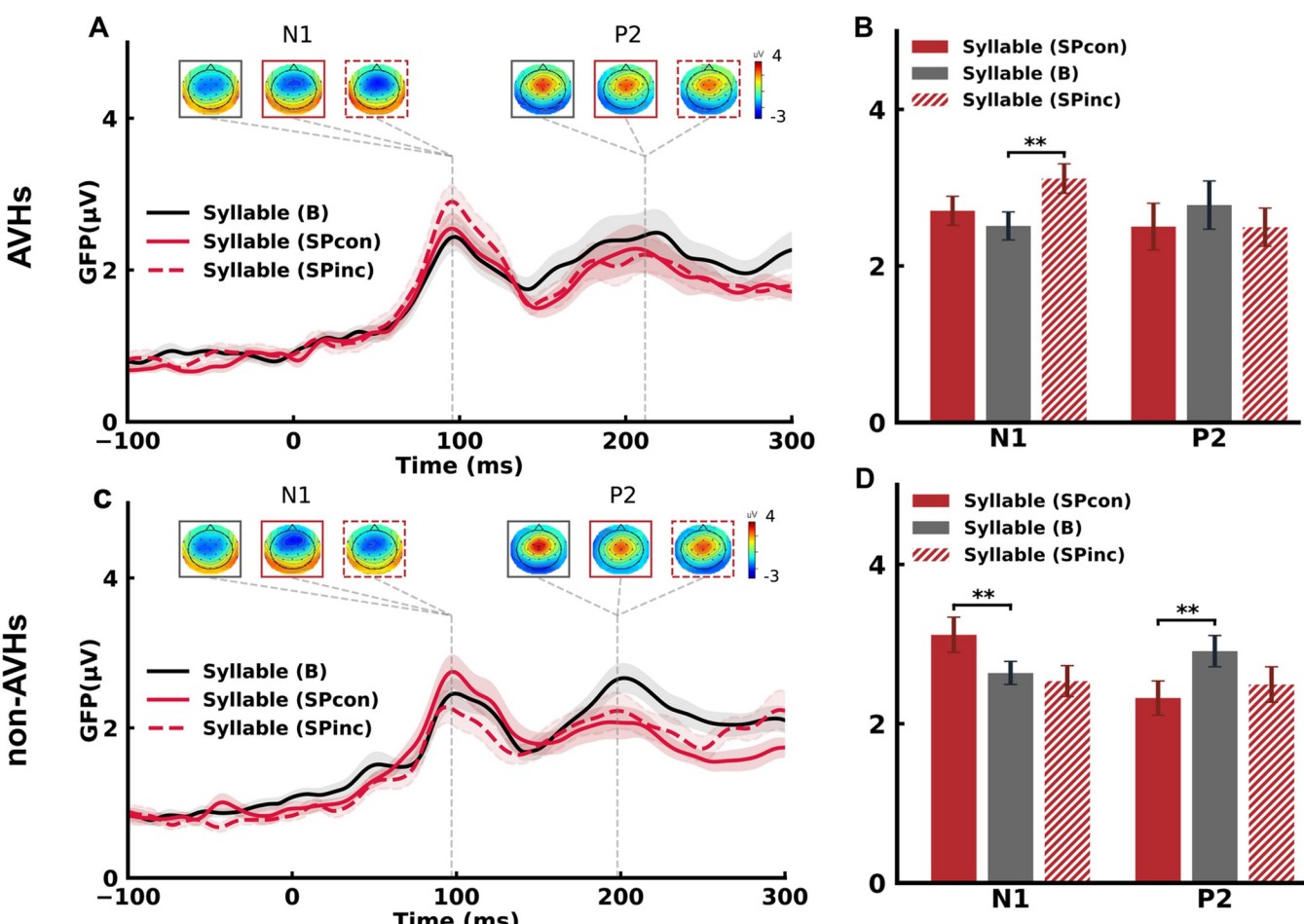

**Fig 4. The opposite modulation effects on auditory responses during SP between AVHs and non-AVHs groups.** (A) ERP time course and topographic responses for SP and B conditions in AVHs patients. Typical N1 and P2 auditory response components were observed in GFP waveforms of each condition. The response topographies at each peak latency are shown in boxes with the same color code for each condition. (B) Mean GFP amplitude at N1 and P2 latencies for SP (red) and B (gray) conditions in AVHs patients. Responses in SPinc were significantly larger than those in B in the N1 component. (C) ERP time course and topographic responses for SP and B conditions in non-AVHs patients. (D) Mean GFP amplitudes at N1 and P2 latencies for SP and B conditions in non-AVH patients. Responses in SPcon were significantly larger than those in B in N1 components, contrasting with the results in AVHs in (B). Error bars indicate ± SEMs. ∗ For $p < 0.05$, ∗∗ for $p < 0.01$, FDR-corrected for multiple comparisons. (The underlying ERP data for this figure can be found at http://osf.io/rsnu4/ and in S2 Data.) AVH, auditory verbal hallucination; ERP, event-related potential; SP, specific preparation.

For responses to tones, paired *t* tests were conducted between SP and B for N1 and P2 separately. The effects were not significant either in N1 ($t(19) = 0.07$, $p = 0.94$, $d = 0.01$) or in P2 ($t(19) = 1.45$, $p = 0.35$, $d = 0.22$) in the AVHs group (S5A and S5B Fig). In non-AVHs group, the effects in N1 ($t(19) = 0.69$, $p = 0.50$, $d = 0.11$) and P2 ($t(19) = 0.50$, $p = 0.62$, $d = 0.10$) were not significant (S5C and S5D Fig). The N1 and P2 topographies showed typical auditory response patterns in both conditions and groups (S5A and S5C Fig). The lack of modulation on tones during specific preparation in both patient groups was consistent with the results in normal controls [39], indicating that EC contained the task-related information in both AVHs and non-AVHs.

## The correlations between clinical symptoms and neural modulation effects

We further investigate the distinct impairment of motor-to-sensory transformation in schizophrenia by exploring the relation between the clinical symptoms and neural measures of

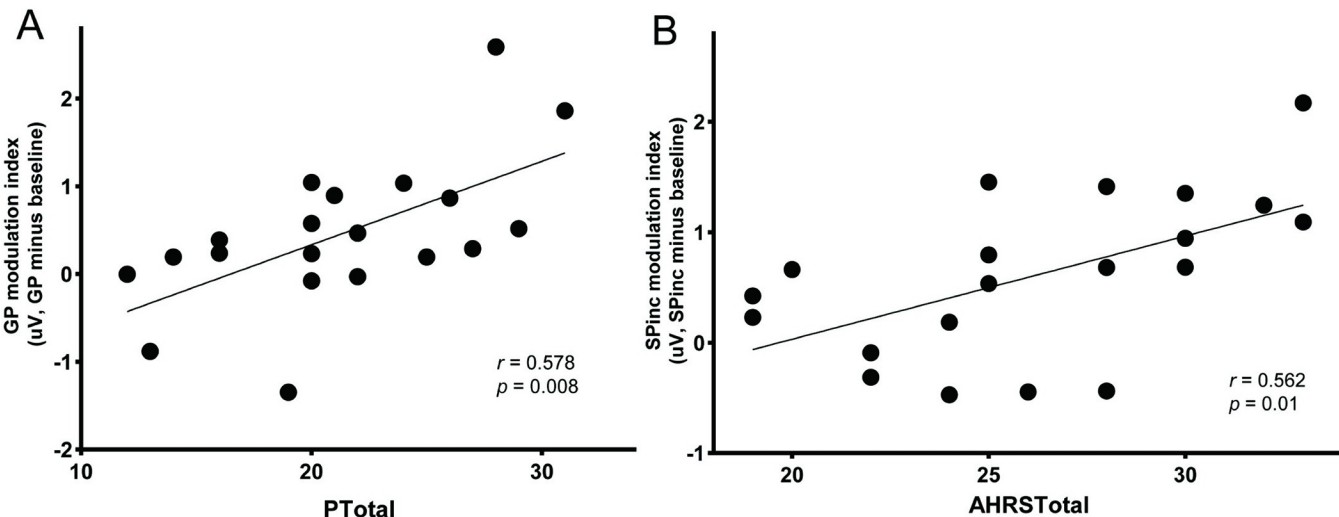

**Fig 5. Correlation between positive symptoms and neural measures in the AVHs group.** (A) The correlation between PTotal scores and GP modulation index. (B) The correlation between AHRS total scores and SPinc modulation index. PTotal represents the positive symptom total scores in PANSS; AHRS total scores represent the severity of auditory hallucinations symptoms; GP modulation index represents the magnitude of the N1 suppression effects in the GP condition; SPinc modulation index represents the magnitude of the N1 enhancement effects in the SPinc condition. (The underlying data for this figure can be found at http://osf.io/rsnu4/ and in S3 Data.) AHRS, Auditory Hallucinations Rating Scale; AVH, auditory verbal hallucination; GP, general preparation; PANSS, Positive and Negative Syndrome Scale.

modulation effects. According to our hypothesis that the CD function is impaired in the AVHs group, a derivation is that the severity of the hallucination symptoms correlates with the degree of impairment in the CD inhibitory function. That is, in terms of neural measures, the severity of the symptoms would correlate with the measures of suppression in the GP condition. Therefore, we carried out Pearson correlation analysis between the GP modulation index (calculated as GP minus B) and 2 positive symptom scores (PTotal: positive symptoms score and AHRSTotal: auditory hallucination symptoms score). A significant positive correlation was observed between the PTotal and GP modulation index after multiple-comparison correction (Fig 5A, $r = 0.578$, $p = 0.008$). These significant symptom-neural correlation results, complementing the predicted negative group level ERP results in the GP condition (Fig 3), provide additional evidence supporting the "broken CD" hypothesis in schizophrenia patients.

According to our hypothesis about EC, the positive symptoms of auditory hallucinations—random perceptual-like experiences without corresponding acoustic stimulations—would be mediated by noisy EC that could activate neighboring auditory neural representations around the sensory target of the action. That is, the noisy EC modulates and enhances the sensitivity of neural responses to unprepared auditory units. To test this hypothesis, a difference score was first calculated by subtracting the N1 response amplitude in the B condition from the SPinc condition in the AVHs group. The difference score represents the magnitude of the N1 enhancement effects in the SPinc condition. The N1 enhancement magnitude significantly and positively correlated with the AHRS total scores (Fig 5B, $r = 0.562$, $p = 0.01$) —the more severe the AVHS symptoms, the more enhancement in the N1 responses in the SPinc condition. These results suggest that the enhancement of N1 response magnitude to the unprepared syllables was related to the degree of AVHs symptom severity—noisy and imprecise EC in AVHs may relate to the various forms of auditory hallucinations.

We further explored the relations between neural measures of CD/EC and negative symptoms. Pearson correlation analyses did not reveal any significant correlation between negative symptom scores and CD/EC metrics in AVHs or non-AVHs groups.

## Modeling results of dissociative impairment of CD and EC in AVHs and non-AVHs

To collaboratively simulate the dissociations of impairment between the effects of CD and EC in AVHs and non-AVHs groups, we quantified our hypotheses in a two-layer neural network model. The upper motor layer had 2 functional pathways that linked to the lower auditory layer: an indirect pathway via an interneuron to inhibit all nodes in the auditory layer, and a direct pathway for modulating the gain of corresponding auditory nodes (Fig 6A). These indirect inhibitory and direct enhancement pathways manifest the functions of CD and EC, respectively. This bifurcation of motor signals successfully explained the distinct modulation directions in GP (Fig 6B, left) and SP (Fig 6C, left) in normal participants [39].

For the non-AVHs group, we set the inhibition strength of the interneuron as a free parameter, while other parameters remained the same as the previous study of fitting the results of the normal population. If only CD at the movement intention stage was impaired but EC at the specific preparation stage was intact in the non-AVHs group, the inhibition strength of the interneuron would be significantly smaller than that of the normal population, whereas the same gain modulation in the direct pathway that used in fitting normal population would be able to fit the results of non-AVHs (Fig 6A, middle). More importantly, we hypothesized a "broken" CD and a "noisy" EC for the AVHs group. The "broken" CD hypothesis led to a similar prediction that the inhibitory strength of the interneuron would be smaller than the normal population, and even smaller than non-AVHs (Fig 6A, right). To test the hypothesis of "noisy" EC, we manipulated the ratio of the gain modulation strength between the prepared and unprepared syllables. The "noisy" EC hypothesis derived a prediction that the best-fitted parameter for the SP results in AVHs would yield relatively more gain over the neighboring auditory nodes than that of the prepared auditory target (Fig 6A, right).

For the non-AVHs group, the simulation results revealed that, during GP, the inhibitory effect was dampened because of deficits in the interneuron. The best-fitted parameter of inhibitory strength was 0.2440, compared with a stronger inhibition of 0.4372 in the normal population. The weaker inhibitory strength made the inhibition of GP disappear (Fig 6B, middle). After temporal averaging of the peak component in the waveform responses, the simulation result of suppression in GP relative to B was consistent with the empirical observations in the *GP* condition for non-AVHs group (Fig 6D). The Bayes factor of comparison between the simulation results and empirical results in *GP* condition favored the null (scaled JZS Bayes factor = 4.30), suggesting the model captured the absence of suppressed auditory responses in the *GP* for non-AVHs group.

For the AVHs group, a similar inhibitory deficit was built in the interneuron (Fig 6A, right). The simulation results suggest that during GP, the inhibitory strength was decreased to 0.0695, much smaller than that of the normal population (0.4372), and smaller than that of the non-AVHs group (0.2440). The much weaker inhibitory strength yielded the absence of inhibition in GP (Fig 6B, right). The difference between temporal averages of the simulated peak components of GP and B in the AVHs group was not different from the empirical results, supported by the Bayes factor (scaled JZS Bayes factor = 4.30), suggesting the model also captured the absence of suppression in the GP for AVH group.

The modulation effects diverged between AVHs and non-AVHs groups in SP—the non-AVHs group had similar effects as the normal population, whereas AVHs had an opposite modulation pattern (empirical results in Fig 4). This was hypothesized as the EC function differences—"noisy" in AVHs group, whereas the EC function remained as precise in the non-AVHs group as normal group. For the simulation of SP results in non-AVHs, the gain modulation function remained the same as that in the normal population, indicated by a similar

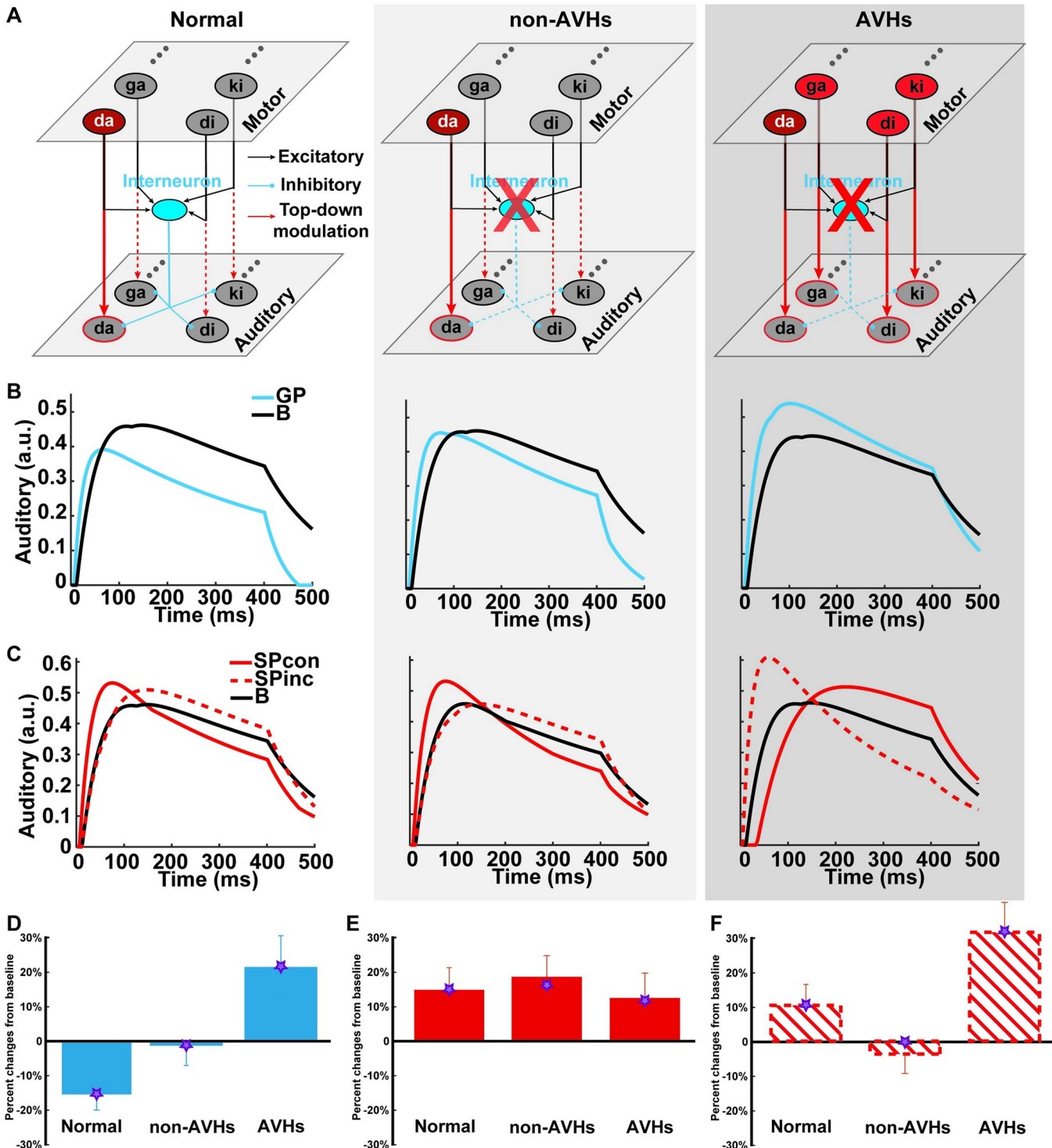

**Fig 6. Model simulation results of distinct impairments of CD and EC in clinical populations.** (A) model architecture and manipulations that quantify the hypothesized neural impairments mediating AVH. Speech units in the motor layer (upper) link to the units in the auditory layer (lower) via 2 pathways. An interneuron receives signals from each unit in the motor layer and inhibits all units in the auditory layer (blue, simulating the inhibitory function of CD during GP). Moreover, signals from each motor unit bifurcate and sensitize its corresponding auditory unit (red arrows, simulating the enhancement function of EC during specific preparation). A trial in which /da/ is prepared to speak is taken as an example, indicated by the only red in the motor "da" neuron with a solid arrow pointing to the auditory "da" neuron with a red circle. Other unprepared syllables do not activate the motor neurons and dashed arrows indicate no modulation on the corresponding auditory neurons. Compared with the normal population (left plot), for non-AVHs (middle plot), the weaker CD function is modeled as the reduced inhibition strength of the interneuron (indicated by a lightened red cross over the interneuron, with dashed blue lines representing

reduced inhibition in downstream). Whereas the modulation strength of the EC enhancement function is preserved in non-AVHs, indicated by the same activation in the motor layer and modulation patterns to auditory neurons as the normal population (left). For AVHs (right plot), the inhibitory function of CD could be even more impaired (a red cross over the interneuron) and the enhancement function of EC is imprecise as its modulation not only on the target unit of "da" but also over all neighboring units (red in all motor units and solid arrows to all auditory units). The simulation results of time course responses in (B) GP and in (C) SP conditions for normal (left), non-AVHs (middle), and AVHs (right) groups. Each line represents the simulated dynamics of responses from the corresponding auditory units, with a black line for the B condition, a blue line for the GP condition; solid and dashed red lines for SPcon and SPinc, respectively. The simulation results of component response magnitude compared with the empirical data in (D) GP, (E) SPcon, and (F) SPinc conditions. The bars represent the observed modulation effects of speech preparation on N1 auditory responses. The stars on the bars represent the simulation results in a given condition and group. Both empirical and simulation results are normalized by baseline conditions. The model simulations capture the absence of inhibition during GP as well as the opposite modulation patterns in SP among different groups. (The underlying data for this figure can be found at http://osf.io/rsnu4/ and in S4 Data.) AVH, auditory verbal hallucination; CD, corollary discharge; EC, efference copy; GP, general preparation.

activity pattern of motor units and of modulation arrows to auditory units between non-AVHs (Fig 6A, middle) and normal (Fig 6A, left). The simulation results showed the enhancement effects in SP compared to B in the non-AVHs group (Fig 6C, middle). The temporal averages around the peak of simulated waveform responses were not statistically different from the empirical data, supported by the Bayes factor (scaled JZS Bayes factor = 4.00 for SPcon and 3.57 for SPinc), suggesting the intact modulation of EC in non-AVHs. Interestingly, the simulation results of SP in non-AVHs must use the normal inhibitory strength of the interneuron rather than the simulated decreased inhibitory strength obtained in GP of non-AVHs and reduce the auditory input connection strength to adjacent auditory nodes by about 30%. This indicated that the inhibitory function may vary across the preparation stages that potentially originated from different hierarchies of the neural pathway for actions [39] and the deficits of CD in the non-AVHs group may be only in the early intentional stage but not in later preparatory stages.

For the AVHs group, in addition to the inhibitory deficits in the interneuron, the gain modulation function was hypothesized to be "noisy." The "noisy" gain modulation was modeled as a parameter of the ratio between the modulation gain on prepared and unprepared auditory nodes. That is, the gain modulation in AVHs may not be as precise as that in normal or non-AVHs groups (indicated in Fig 6A right plot, the motor units for the unprepared syllables were also red and the downstream modulations were all solid arrows in AVHs, compared to the inactivated motor units and dashed downstream modulation arrows for the unprepared syllables in the normal and non-AVHs group in Fig 6A, left and middle). The simulation was performed simultaneously for the SPcon and SPinc results in AVHs. The best-fitted parameter of the ratio was 2.848, yielding the modulation gain between prepared and unprepared nodes of 0.351:2.848, compared with that in normal and non-AVHs groups of 1.919:1. This more "spread-out" modulatory gain to the neighboring nodes of the prepared target resulted in smaller responses to SPcon but stronger responses to SPinc as compared to B (Fig 6C). The temporal averages around the peak of simulated waveform responses were statistically not different from the empirical data, as supported by the Bayes factor (scaled JZS Bayes factor = 4.27 for SPcon, and 4.30 for SPinc), suggesting the imprecise modulation by a "noisy" EC in AVHs.

## Discussion

We investigated the modulation functions of motor signals on auditory processing at distinct stages of speech preparation in schizophrenia patients. Our behavioral, electrophysiological, and modeling results collaboratively demonstrated distinct impairments in the motor-to-sensory transformation between subgroups of clinical populations. The symptoms of AVHs are mediated by the impaired source monitoring function of *CD* that results in the absence of inhibition of auditory responses in the general speech preparation, as well as the imprecise activation function of *EC* that results in the varied enhancement and sensitization of auditory cortex

during specific speech preparation. These results suggest that "broken" *CD* plus "noisy" *EC* causes erroneous monitoring of the imprecise generation of internal auditory representation and hence yields AVHs.

The absence of suppression effects during GP in schizophrenia patients is the negative evidence supporting that *CD* signals can be generated during the movement intention stage even without any preparatory contents. Compared with the function of ubiquitously suppressing neural responses to speech in normal controls [39, 40], the *CD* did not function in both groups of schizophrenia patients with and without AVHs at the earliest stage of motor intention (Fig 3). These results were consistent with immense literature about the weaker or absent action-induced sensory suppression in schizophrenia [11,27,41–43], as well as findings that schizophrenia patients elicit a smaller readiness potential before movement [44] than normal controls [41,45]. Our results delineate the temporal dynamics of the impaired function of *CD* that can occur in the earliest stage of motor intention, complementing with and extending from most findings during the action execution.

Different causes may mediate the absence of inhibition effects during GP in distinct clinical subgroups. Lack of desire to act may be a common cause in both patient groups that decreases the inhibitory strength of *CD* on auditory processes (Fig 3) because no significant differences were found in the severity of their negative symptoms. The ubiquitous pathophysiological negative symptoms in schizophrenia may generate weaker *CD* in the motor intention stage in both patient groups. That is, the lack of inhibitory effects during general action preparation may be a biomarker for less intention and negative symptoms. However, impaired *CD* functions in motor-to-sensory transformation may be a cause that is unique in AVHs in addition to the deficits in the generation of *CD* due to negative symptoms.

Although as predicted, no difference in the negative empirical results of modulation by GP (Fig 3), the symptom-neural correlation revealed a significant correlation between the severity of positive symptoms and impairment of CD inhibitory function in AVHs group, providing strong evidence at the individual level supporting the hypothesis (Fig 5A). Moreover, computational model results (Fig 6) reveal (1) a greater degree of impairment in the inhibition strength of the interneuron in AVHs compared with non-AVHs; and (2) the impaired function of *CD* continued throughout the specific speech preparation stage in AVHs, whereas the empirical results of modulation in SP in non-AVHs require the intact *CD* inhibitory strength to fit. All these pieces of evidence regarding the degree and temporal extent of the inhibitory function deficits suggest the impairment of *CD* in the motor-to-sensory transformation in AVHs. Our results of more severe impairment of *CD* in GP and throughout specific preparation in AVHs but not in non-AVHs are consistent with the findings that transcranial magnetic stimulation affected the sense of agency only when stimulation time locked in action planning, rather than in the physical consequences of the actions appeared [46]. The empirical and modeling results consistently support that the impairment of *CD* function is associated with an abnormal sense of agency in AVHs patients [28].

The *EC* function and its impairment also show dissociation between AVHs and non-AVHs. In non-AVHs patients, the *EC* function is the same as normal—the motor signals in specific preparation enhanced the neural responses only to the prepared syllable (Fig 4). However, in AVHs patients, the motor signals in specific preparation enhanced the neural responses to the unprepared syllable (Fig 4). And the modulation effects of enhancement are positively correlated with the severity of AVH symptoms (Fig 5B). Modeling results further quantified that the different modulation patterns between AVHs and non-AVHs were caused by the imprecise modulation from the motor to sensory units that provide incorrect gains over the non-target of the specific preparation (Fig 6). These results suggest that *EC* can be generated in the motor-to-sensory transformation pathway during specific preparation in AVHs

patients. However, the EC is "noisy" either in the generation process in the motor system or it is imprecisely mapped onto the auditory system. As a result, the "noisy" EC modulates and enhances the sensitivity of neural responses to unprepared auditory units. The empirical ERP modulation effects, correlation results with AVH symptoms, and model simulation results collaboratively support the hypothesis of "noisy" EC in AVHs. This imprecise EC in AVHs may relate to the "non-sense" and various forms of auditory hallucinations.

In previous studies, action-induced suppression [21,22,47,48] and enhancement [30–34] have been observed in the normal population and animal models. By considering the different characteristics of motor signals across temporal dynamics of motor processes, it has been proposed that the copy of motor signals at different action stages may distinguish into *CD* and *EC* that mediate distinct functions for regulating actions [39]. In this study, we found distinct impairments in AVHs between general and specific preparation stages (Figs 3 and 4). The observed distinct impairments in different speech preparation stages offer evidence from a clinical perspective that is consistent with empirical neuroscience results in the normal population and supports the updated theoretical framework of internal forward models [39].

The observed double dissociations of *CD* and *EC* functions between AVHs and non-AVHs reveal the impairments in the motor-to-sensory transformation that mediate the positive symptoms of auditory hallucinations. The positive nature of auditory hallucinations requires the active construction of neural representations that mediate perceptual-like experience. Our observation of "noisy" *EC* that imprecisely sensitizes auditory cortices provides a foundation for inducing subjective experience without external stimulation. Together with the impairment in *CD* that leads to less suppression and hence deficits in labeling the sources that induce neural responses, hallucinations about experiencing perceptual events would occur. That is, the combination of impairments on distinct functions between motor and sensory systems mediate the positive symptoms of auditory hallucination, which is consistent with the hypothesis of motor-to-sensory transformation as an origin of hallucinations [16,27,49–51]— the impaired monitoring function misattributes the sources of internally motor-induced [8] or other top-down–induced neural representations [52,53]. The conceptually, anatomically, and functionally distinct motor signals of *CD* and *EC*, instead of sole inhibitory function in the motor-to-sensory transformation, collaboratively contribute to the positive symptoms of auditory hallucinations.

Most previous studies have explored the differences in impaired motor-to-sensory transformation signals between schizophrenia patients and normal controls [28,54–56]. In this study, we explored how the motor signals regulated perceptual neural responses between subgroups of schizophrenia patients with different symptoms. This hypothesis-driven symptom-based approach yields novel insights into the potential neural mechanisms that mediate different aspects of deficits in schizophrenia. By distinguishing the uniqueness of psychotic symptoms in the same categorized mental disorder, the distinct impairments in the motor-to-sensory transformation have been revealed, which delineates the potential deficits mediating positive symptoms in mental disorders. Moreover, considering the overlapping symptoms between subgroups of patients can provide insights into possible causes, for example, the negative symptoms in both AVHs and non-AVHs may hint at some common deficits in the motor system mediating similar negative symptoms in anhedonia and amotivational syndrome. The approach of comparing unique and common symptoms may expand over different types of mental disorders, such as auditory hallucinations in schizophrenia and bipolar disorder to investigate the potential common causes from a cognitive neuroscience perspective. Adding the neural bases of symptoms across different types of mental disorders complements the symptoms-based categorization and may provide a 2D matrix for a more precise diagnosis of mental disorders [57].

Our study highlights cognitive computation as a crucial interface to bridge neural circuits to mind and behavior, especially in understanding mental disorders. Recently, computational psychiatry has emerged as a novel quantitative cognitive account for probing the mechanisms that mediate mental disorders [58–61]. In this study, we utilized a computational modeling approach and identified the subtle differences in the negative results during GP between AVHs and non-AVHs. Moreover, the computational modeling enables us to differentiate the distinct impairments of *CD* and *EC* throughout the evaluation of actions—parametric simulation using hypothesized intact and impaired values overcome the temporal overlaps of *CD* and *EC* in the specific preparation stage that would be hard, if not impossible to investigate using behavioral or noninvasive cognitive neuroscience approaches on human participants with mental disorders. The consistent EEG and modeling results mechanistically reveal the predictive functions of motor and sensory networks that may mediate the symptoms of psychosis [16,38]. Our endeavors of combining behavior, electrophysiology, and modeling manifest Marr's computational approach [62] and provide a possible link between mental and behavioral status with neural circuits [63,64]. The computational approach puts the cognition back to the investigation of mental disorders [65] and yields testable hypotheses at the cognitive, system, and even cellular and molecular levels to collaboratively understand mental disorders.

Our neural network model aligns with previous Bayesian inference models that explain the positive symptoms of schizophrenia. False prediction errors have been assumed as the cause of the positive symptoms [38,66,67]. The inaccurate prior expectancies distort the integration with the sensory evidence, and the resultant false prediction errors create disturbance in inference updating. Specific to hallucinations, recent models propose that abnormally strong priors dominate and overwrite prediction errors so that both contents and the presence of auditory hallucinations can be explained [68]. Our model, based on 2 distinct modulatory functions in the motor-to-sensory transformation, assumes 2 types of predictions that can presumably integrate both inference models. EC serves as a content prediction, resembling the strong priors that would elicit the specific auditory neural representation without corresponding external stimulation [4,8,35,36]. Whereas CD serves as a prediction for the sense of agency, and the impairment would cause the failure to explain away the internally generated auditory neural representations that are evoked by EC [40,49,52,69]. Moreover, our neural network model provides a mechanistic account and potentially ground the psychological constructs of inference and prediction in the motor and sensory systems. The hypothesized origins of auditory hallucinations at the level of neural representation and computation can inspire future neuroscience studies at the cellular, system, and cognitive levels.

By probing the impairments in the interactive neural processes between motor and sensory systems, we observed the functional distinctions between *CD* and *EC*, and their impairments in relation to auditory hallucinations in schizophrenia. The pathophysiology of schizophrenia involves a ubiquitously distributed motor-sensory circuitry in which the "broken" *CD* dysfunctionally misattributes the sources of the neural activity induced by "noisy" EC—the failure of dampening the internally induced sensory neural activity leads to hallucinatory experiences. Distinct impairments in functional granularity of motor-to-sensory transformation mediate positivity symptoms of agency deficits in mental disorders.

## Materials and methods

### Participants

Twenty patients (10 males), who matched the DSM-V diagnosis of schizophrenia and were experiencing AVHs without hallucinations in other modalities (AVHs group), were recruited from Shanghai Mental Health Center. Another group of 20 patients (14 males), who met a

DSM-V diagnosis of schizophrenia and had never experienced AVHs were recruited from the same hospital (non-AVHs group). Two experienced psychiatrists independently confirmed the diagnoses based on the Structured Clinical Interview for DSM-V. All patients were right-handed with an age range of 18 to 45 years old. They received antipsychotic medications and were clinically stable during the experiment.

In this study, the non-AVHs group is defined at the trait level—the non-AVHs patients have never experienced AVHs throughout the course of their illness. Evaluation criteria were based on (1) Comprehensive Clinical Interviews: Detailed interviews with patients, covering their entire psychiatric history, to confirm the absence of AVHs. (2) Patient's subjective report: By clearly defining what constitutes AVHs and making sure that patients understand the symptoms of AVHs, and report that they have never experienced AVHs. (3) Medical Records Review: Examination of medical records over the course of the illness to ensure no documented episodes of AVHs. (4) Collateral Information: Gathering information from family members or caregivers who have been consistently involved in the patient's care to corroborate the absence of AVHs. These strict criteria were used to better control and contrast with the AVHs group to test our hypothesis.

The study was approved by the Institutional Review Board at New York University Shanghai and the Institutional Ethics Committee at Shanghai Mental Health Center. Patients provided written informed consent before they participated in the study. This study was performed in accordance with the guidelines laid out in the Helsinki Declaration of 1975, as revised in 2008.

Previous results [39] of 19 normal participants who completed the same GP task as the patients and 16 normal participants who completed the same SP task served as a normal control group. The result figures of the normal control group were attached to the supplementary materials.

## Clinical measures

All participants' demographic data were collected and reported. The duration of illness of each patient was recorded. Symptom rating interviews were done on the day of EEG recordings. Trained psychiatrists implemented a one-time cross-sectional psychotic symptoms assessment at individual enrollment using the PANSS [70]. The PANSS includes 30 items that measure the presence and severity of positive, negative, and general symptoms on a seven-point scale ranging from 1 to 7. Hallucination severity was rated from the P3 subscore of the PANSS, with the higher rating indicating an increased hallucination severity. Non-AVHs patients had a rating of 1 in the P3 subscore, indicating the hallucination symptom was absent. Furthermore, the severity level of AVHs was assessed using the seven-item Auditory Hallucinations Rating Scale (AHRS) [71].

All enrolled patients were treated with 1 or 2 second-generation oral antipsychotic drugs (e.g., olanzapine, risperidone, aripiprazole, clozapine, quetiapine, and amisulpride). We chose olanzapine as the standard drug, for every patient, the cumulative antipsychotic dose was transformed into the equivalent dose to olanzapine based on defined daily doses (DDDs) presented by the WHO Collaborative Center for Drug Statistics Methodology [72]. Statistical analysis showed that there was no significant difference in the mean dosage between the AVHs and non-AVHs groups ($t(1,38) = 1.501$, $p = 0.142$).

## Materials

Four auditory syllables (/ba/, /pa/, /ga/, /ka/) with a duration of 400 ms were synthesized via the Neospeech engine (www.neospeech.com) at a 44.1 kHz sampling rate in a male voice.

Moreover, a 1 k Hz pure tone with the same duration of 400 ms was included in the experiment. All auditory stimuli were delivered binaurally at 70 dB SPL through plastic air tubes connected to foam earplugs (ER-3C Insert Earphones; Etymotic Research). A Shure Beta 58A microphone was used to detect and record participants' vocalization.

## Procedures

We first provide an overview of the procedures. Two tasks were used to elicit different speech preparation stages before articulation. This design provided the temporal segregation of motor signals and induced CD and EC in different stages of motor preparation. During each preparation stage, auditory probes were introduced to explore how distinct preparation stages (and hence different motor signals of CD and EC) modulate perceptual responses to auditory stimuli. Below we describe the details of the procedures.

Fig 2A illustrates the trials of the GP task. The trial started with a cross fixation displayed for 500 ms. A yellow visual cue of 2 symbols (#%) then appeared in the center of the screen for a duration in a range between 1,500 ms and 2,000 ms with an increment of 100 ms. Participants were asked to prepare to speak in the upcoming articulation task, although they did not know what to say because the symbols did not contain any linguistic information. In half of the trials, an auditory probe, either one of the 4 auditory syllables (/ba/, /pa/, /ga/, and /ka/) or a 1 k Hz pure tone, was played during the last 400 ms of the preparatory stage. Another half of the trials did not include auditory probes (GP$_{NS}$). The mixed-trial design aimed to enforce participants' preparation for the upcoming articulation task based on visual cues instead of auditory probes. After a blank screen of a range of 200 ms to 400 ms with an increment of 50 ms, participants saw a green visual cue that was one of the 4 syllables (/ba/, /pa/, /ga/, and /ka/) in the center of the screen for a maximum of 1,200 ms and were asked to produce the syllable as fast and accurately as possible. The onset time of vocal response was recorded to quantify the reaction time (RT).

Fig 2B illustrates the trials of the specific preparation (SP) task. The procedure was similar to the GP task except for 2 differences. One was that the visual cue during the preparatory stage was a red syllable randomly selected from the 4 syllables (/ba/, /pa/, /ga/, and /ka/). Participants prepared to speak the syllables because the upcoming articulation task was the same syllable with a speeded requirement in a time-limited setting. The other difference was that an auditory probe was presented in every trial during the preparatory stage. The auditory probes were either the same as or different from the visual cue, yielding 2 conditions—auditory syllables were congruent (SPcon) or incongruent (SPinc) with the visual cue (and hence the syllable that participants prepared to speak).

Furthermore, 2 additional types of trials were included to reduce expectations and to evaluate the effects of preparation. In one type of trial, the green visual cue of articulation immediately appeared after the fixation, and participants were asked to articulate the syllable without preparation (NP trial). The RTs in NP trials were used as a behavioral baseline of syllable production speed and were compared with the RTs in preparation trials to quantify the effects of preparation behaviorally. In another type of trial, a white visual cue of symbols (**) appeared after the fixation with auditory probes played during the last 400 ms of the visual cue presentation duration. No articulation stage followed. Participants were only required to listen to the auditory probes passively (baseline, B trial). The B trials contained similar visual cues and auditory probes as those in the GP and SP trials but without preparation, yielding baseline auditory responses for quantifying the neural modulation effects of preparation. The neural responses to the auditory probes in the B trials were compared to those in the GP and SP trials to quantify the modulation effects of different motor signals. The NP and B trials were equally divided into GP and SP blocks to reduce the duration of the experiment.

Therefore, 4 types of trials (NP, GP, GP$_{NS}$, and B) were randomly presented within 6 GP blocks. Each block included 48 trials, yielding a total of 288 trials (96 trials for GP$_{NS}$ and GP; 48 trials for NP and B). Half of the GP trials and half of the B trials contained auditory probes of syllables and another half contained auditory probes of pure tone. The time limit for articulation was set to 1,200 ms for GP and GP$_{NS}$ trials, respectively. The time limit setup was aimed to eliminate expectations and enforce preparation.

Three types of trials (NP, SP, and B) were randomly presented within 5 SP blocks. Each block included 48 trials, yielding 240 trials (144 trials for SP; 48 trials for NP and B). A third of the SP trials were SPcon condition, another third was SPinc condition, and the last third of trials had auditory probes of pure tone. The 48 B trials also contained half trials of auditory syllables and half trials of pure tone. Together with the B trials in GP blocks, a total of 48 trials with auditory syllables and 48 trials with pure tone for the B condition. The time limit for articulation was set to 1,500 ms for NP and 1,000 ms for SP, respectively.

## Demographic and clinical data

Statistical analyses were performed using IBM SPSS (Statistics version 17.0) and GraphPad. Prism 5.02. The normality of data was tested using the Kolmogorov–Smirnov tests. Demographic and continuous variables were subject to one-way ANOVA and Fisher's LSD (least significant difference) post hoc multiple comparison tests ($\alpha = 0.05$), whereas the categorical values were subject to the chi-squares test. Data were presented as mean and standard deviation. Effects at $p < 0.05$ were considered significant.

## Behavioral data analysis

The RTs of the articulation were calculated as the time lag between the onset of the green visual cue and the participants' vocalization. In AVHs and non-AVHs groups, the averaged RTs were obtained in each of the 4 trial types (NP, GP, GP$_{NS}$, and SP). The RTs were subject to a repeated-measures one-way ANOVA and a post hoc Tukey Student $t$ test for pairwise comparisons.

## EEG data acquisition and preprocessing

Neural responses were recorded using a 32-channel active electrode system (Brain Vision acti-CHamp; Brain Products) with a 1,000 Hz sampling rate in an electromagnetically shielded and sound-proof room. Electrodes were placed on an EasyCap, on which electrode holders were arranged according to the 10 to 20 international electrode system. The impedance of each electrode was kept below 10 kΩ. The data were referenced online to the electrode of Cz and re-referenced offline to the grand average of all electrodes. Two additional EOG electrodes (horizontal: HEOG; vertical: VEOG) were attached for monitoring ocular activity. The EEG data were acquired with Brain Vision PyCoder software (http://www.brainvision.com/pycorder.html) and filtered online between DC and 200 Hz with a notch filter at 50 Hz.

EEG data processing and analysis were conducted with customized Python codes, MNE-python (https://github.com/mne-tools/mne-python) [73], Autoreject (https://github.com/repos/autoreject) [74], EasyEEG (https://github.com/ray306/EasyEEG) [75], and the Topography-based Temporal-analysis Toolbox (TTT, https://github.com/TTT-EEG/TTT-EEG) [76]. For each participant's data set, bad channels were replaced with the average of their neighboring channels. The data set was band-pass filtered with cutoff frequencies set to 0.1 and 30 Hz. The filtered data set was then segmented into epochs ranging from −200 ms to 800 ms, relative to the onset of the auditory probe, and baseline corrected using the 200 ms pre-stimulus

period, and 240 epochs in the GP task and 192 epochs in the SP task were extracted for each participant.

During EEG recording, we attached 2 electrodes (EOG) near the eyes to monitor the vertical and horizontal eye muscle movement. In EEG preprocessing, we utilized the 2 EOG and 32 EEG channels and created epochs based on EOG events (eye blinks or movements) and EEG events in the data set. Initially, epochs containing conspicuous artifacts due to eye blinks or movement (generally >100 μv) were manually excluded. Subsequently, we employed the Autoreject toolbox, a Python package designed for automated artifact rejection. This tool explicitly marks the EOG channels and correlates the timing of activation in the EOG channels with the EEG channels and calculates an optimal rejection threshold for each channel using the "get_rejection_threshold" function. This function assesses the signal characteristics of individual channels to determine appropriate peak-to-peak amplitude thresholds. Epochs exceeding these calculated thresholds were automatically rejected, ensuring our analysis was confined to data free of significant ocular artifacts.

In the AVHs group, for trials with auditory syllable probes, on average, 32, 37, 39, and 37 trials were included in B, GP, SPcon, and SPinc, respectively. For trials with auditory probe of tones, on average, 35, 38, and 37 trials were included in B, GP, and SP, respectively. In the non-AVHs group, for trials with auditory syllable probes, on average, 36, 38, 40, and 39 trials were included in B, GP, SPcon, and SPinc, respectively. For trials with auditory probes of tones, on average, 36, 39, and 41 trials were included in B, GP, and SP, respectively. The ratio of trial rejection in the AVHs group and the non-AVHs group was 23.83% and 18.75%, respectively.

Event-related potentials (ERPs) to the auditory probes were obtained by averaging trials in each condition (syllables in GP, SPcon, SPinc, and B; tones in GP, SP, and B) for each participant in the AVHs and non-AVHs groups. The global field power (GFP)—the normalized geometric mean across 32 electrodes—was calculated from each ERP. The GFP represents the overall response power changes over time, which is an optimal measure in a novel study to balance the requirements of exploration and to overcome problems of false positives by avoiding subjective channel selections, multiple comparisons, and individual differences [77,78]. Individual peak amplitudes and peak latencies of the N1 and P2 components in the GFP waveforms were automatically identified using the TTT toolbox in predetermined time windows of 50 to 150 ms and 150 to 250 ms, respectively [76]. We visually verified whether the identified peaks were correct in each participant.

## EEG data analysis

The identified ERP component responses were used in the following statistical tests. First, within-group analyses were performed, separately in the AVHs and non-AVHs groups, to test the hypotheses about the impairment of motor signals and their modulation effects in each group. Paired $t$ tests were carried out on the GFP responses to syllables in the hypothesis-driven paired comparisons (GP and B, SPcon and B, and SPinc and B). Repeated measures one-way ANOVAs were conducted on GFP responses to tones (GP, SP, and B). Both statistical tests were performed separately for the N1 and P2 components. Furthermore, two-way mixed ANOVAs were conducted to test the between-group differences, separately for the GP, SPcon, and SPinc conditions. The factor of the group (AVHs, non-AVHs, and normal) was set as a between-subject factor. The condition (GP and B, SPcon and B, and SPinc and B) was set as the within-subject factor. For ANOVAs, effect sizes were indexed by partial $\eta^2$. For paired $t$ tests, effect sizes were indexed by Cohen's d. Significant effects were determined by $p < 0.05$ and partial $\eta^2 > 0.14$ [79].

## Bootstrap simulation analysis

Because of the costs and efforts to conduct a direct replication of the observed null results in the GP condition, we implemented a bootstrapping procedure to estimate the reliability of the null results. Because the empirical sample of 20 participants was randomly drawn from the population, it should be representative of the population. Therefore, by resampling from the empirical sample many times with replacement, it would be as if many samples were drawn from the original population, and the estimates derived would be representative of the theoretical distribution [80,81]. Specifically, we carried out the bootstrap analysis on the N1 component data of the GP condition in the AVH group where we predicted a null result derived from the "broken CD" hypothesis. We first subtracted the baseline responses from the responses of the GP condition, yielding the GP modulation index for each individual. Then, we performed 10,000 times resampling with replacement and generated an estimated GP modulation in the AVH population. The confidence interval was obtained by taking the range of 99% variance of the distribution.

## Modeling

We adapted the model in [39] to quantitatively test our hypotheses regarding the deficits of CD and EC in AVHs and non-AVHs groups. Specifically, 2 free parameters were introduced in the model to adjust the values that represent the CD and EC functions. The first parameter was *the inhibitory strength* of an interneuron to simulate the hypothesis of impaired CD during GP. The second parameter was *the ratio of modulation gain* between prepared and unprepared auditory tokens to simulate the hypothesis of "noisy" EC during SP in AVHs. Next, we described the models and the incorporation of 2 parameters to test our hypotheses.

To quantify the proposed double dissociation between the impairment of CD and EC in AVHs and non-AVHs groups, we built a two-layer neural network model to simulate the dynamics and modulation effects of motor signals on sensory processing (Fig 6A). The model structures are similar to the previous model of simulating CD and EC in the normal participants [39] (Fig 6A, left plot). The upper layer represents motor processing, and the lower layer denotes auditory processing. Each layer includes multiple neurons that represent different syllables. Each neuron in the auditory layer is a rate-coded unit with synaptic depression. The updating of membrane potential is governed by Eq (1).

$$\frac{dv_i(t)}{dt} = \tau\left\{g_i'(1-v_i)\sum_j w_{ij}e_j - v_i[L + I(\sum_k o_k + n*m')]\right\} \tag{1}$$

The membrane potential of an auditory neuron, $v_i$, is updated according to the sum of 3 sources. The first source is an excitatory input from acoustic signals, $e_j$, via bottom-up connections with connection strength, $w_{ij}$. This bottom-up input drives the membrane potential to 1 (governed by the multiplier of 1–$v$). The second source is the leak with the fixed term, $L$. The third source is the inhibition that results in the multiplication of inhibition strength, $I$, and a sum of 2 terms. One term is lateral inhibition which is the sum of output at time $t$ from $k$ units at the auditory layer. Another term is the inhibition from the motor layer, $n^*m'$, which is specified next. The combination of the leak and inhibition drives the membrane potential toward 0 (as the term in the bracket is multiplied by–$v$). The updating speed is proportional to the integration rate (time constant, $\tau$). The sum of 3 sources multiplied by the integration rate yields the updating magnitude for the membrane potential at each time. The fixed parameters are similar to those used in the previous study [39].

The influences of motor signals are modeled as 2 sets of free parameters. The motor signals come from the same motor units but are split into 2 sources. One source integrates activities of

all motor units into an interneuron that inhibits all auditory neurons (Fig 6A). For simplification, the inhibition effect of each motor neuron is assigned as a unit value, $m$. The equivalent inhibition effects from the interneuron are the sum of $n$ motor units, $n^*m$. This motor source simulates the hypothesized function of CD. Another source directly modulates corresponding auditory neurons. This motor signal is modeled as a gain control parameter, $g_i$, which increases the gain of excitatory input to the corresponding auditory unit. This motor source simulates the hypothesized function of EC.

The previous study [39] combined the implementation of an interneuron and gain modulation in one neural network model to collaboratively simulate the inhibition and enhancement throughout the time course of speech production in normal participants. In this study, we manipulated these 2 key parameters of CD (the parameter of $m$) and SP (the parameter of $g_i$) to simulate the results of AVHs and non-AVHs groups. Specifically, during the simulation of GP, because of the hypothesis of CD impairment, we fitted the model to the observed suppression during GP by adjusting the parameter of inhibition strength, $m'$, separately for AVHs and non-AVHs group (the apostrophe after the label of the parameter indicates changes from the value obtained from the normal group).

During the simulation of SP, only the prepared syllable in the motor layer is activated in normal participants. We hypothesized that the non-AVHs group would have intact EC function as normal participants, but the AVHs group would have a "noisy" EC function. We fitted the model to the observed reversal of SP effects in AVHs by adjusting the parameter of $g'_i$. Specifically, for simulation of the AVHs group, the parameter of gain modulation, $g'_i$, was modified proportionally according to the prepared versus unprepared syllables—the gain from the prepared motor unit was decreased by X times, at the same time the gain from the unprepared motor units were increased by X times. That is, compared with normal participants and the non-AVHs group, the signal-to-noise ratio (i.e., the ratio between the gain to the prepared versus unprepared auditory nodes) was adjusted.

To assess how the model fitted the empirical results, the model was validated by fitting the simulated dynamic responses from the auditory output to the empirical EEG data. The fitting was at the level of overall measures—temporal averages of the simulated neural dynamic responses (analogs to the procedure of obtaining the ERP components) and compared to the modulation effects in auditory ERP responses. We treated the model simulation results as the mean from a distribution with unknown variance. Empirical ERP N1 component data in GP and SP conditions were subject to one-sample $t$ tests against the temporal averaged simulation results, separately for non-AVHs and AVHs groups. Because this analysis was to test the null hypothesis that the model simulation results were from a similar distribution of empirical results, we used a Bayesian analysis method for one-sample $t$ tests [82] (online tool at http://rdrr.io/rforge/BayesFactor/man/ttest.tstat.html). The Bayes factor is B01 = M0/M1, where M0 and M1 are the marginal likelihood for the null and alternative, respectively. That is, the Bayes factor is an odds ratio between the null and alternative hypotheses, which indicates that the null is B01 times more probable than the alternative. The parameters for the Bayesian analysis were a sample size of 20 and 20 for AVHs and non-AVHs groups and a scale $r$ on an effect size of 0.707.

## Supporting information

**S1 Fig. The modulation effects on auditory responses during general preparation (GP) in the normal control group.** (A) ERP time course and topographic responses for GP and B conditions in normal populations. (B) Mean GFP amplitude at N1 and P2 latencies for GP (blue) and B (gray) conditions in normal populations (adapted from [39]). In the normal population,

the amplitude of early N1 response in GP was less than that in B ($t(18)$ = 3.406, $p$ = 0.003, $d$ = 0.3). The amplitude of the later P2 response in GP was reduced relative to B ($t(18)$ = 2.240, $p$ = 0.038, $d$ = 0.342). These results suggested that CD that was induced during GP suppressed auditory responses. (This Figure reproduced with permission from the original publisher [39] and Oxford University Press (license number 5858081359804).) The underlying data for this figure can be found at http://osf.io/rsnu4/ and in S2 Data.
(JPEG)

**S2 Fig. Bootstrap analysis of resampling and estimating confidence interval.** Bootstrapping results of N1 component in GP condition of AVH group. The simulation is done by resampling with replacement 10,000 times in the GP modulation index of the empirical sample (GP minus baseline). The blue vertical line denotes the mean of the empirical sample. The horizontal red bar indicates the 99% confidence interval (CI). The suppression (modulation index value less than 0) is outside the CI. The underlying data for this figure can be found at http://osf.io/rsnu4/ and in S2 Data.
(JPG)

**S3 Fig. The absence of modulation effects on tones during general preparation.** (A) ERP time course and topographic responses for GP and B conditions in AVHs patients. Typical N1 and P2 components were observed in the GFP waveforms for each condition. The response topographies at each peak latency are shown in boxes with the same color code of conditions. (B) Mean GFP amplitude at N1 and P2 latencies for GP (blue) and B (gray) conditions in AVHs patients. (C) ERP time course and topographic responses for GP and B conditions in non-AVHs patients. (D) Mean GFP amplitudes at the N1 and P2 latencies for GP (blue) and B (gray) conditions in non-AVHs patients. No significant differences between GP and B were observed in either group. Error bars indicate ± SEMs. The underlying data for this figure can be found at http://osf.io/rsnu4/ and in S2 Data.
(TIF)

**S4 Fig. The modulation effects on auditory responses during specific preparation (SP) in the normal control group.** (A) ERP time course and topographic responses for SP and B conditions in non-AVHs patients. (B) Mean GFP amplitudes at N1 and P2 latencies for SP and B conditions. Responses in SPcon were significantly larger than those in B in N1 components (adapted from [39]). In the SP, early neural responses of N1 were larger than that in B when the auditory syllables were congruent with the SP visual cues (SPcon) ($t(15)$ = −2.49, $p$ = 0.025, $d$ = −0.432). The effect was not significant in the later auditory responses of P2 ($t(15)$ = 0.248, $p$ = 0.808, $d$ = 0.039). However, when the auditory syllables were incongruent with the specific preparation (SPinc), the effect in N1was not significant ($t(15)$ = −1.48, $p$ = 0.160, $d$ = −0.283), nor in P2 ($t(15)$ = 2.024, $p$ = 0.061, $d$ = 0.342). These results suggested that motor signals during SP modulated the perceptual responses based on the content congruency. (This figure reproduced with permission from the original publisher [39] and Oxford University Press (license number 5858081359804).) The underlying data for this figure can be found at http://osf.io/rsnu4/ and in S2 Data.
(JPEG)

**S5 Fig. The absence of modulation effects on tones during specific preparation.** (A) ERP time course and topographic responses for SP and B conditions in AVHs patients. Typical N1 and P2 components were observed in the GFP waveforms of each condition. The response topographies at each peak latency are shown in boxes with the same color code of conditions. (B) Mean GFP amplitude at N1 and P2 latencies for SP (red) and B (gray) conditions in AVHs patients. (C) ERP time course and topographic responses for SP and B conditions in non-

AVHs patients. (D) Mean GFP amplitudes at the N1 and P2 latencies for SP (red) and B (gray) conditions in non-AVHs patients. No significant differences between SP and B were observed in either group. Error bars indicate ± SEMs. The underlying data for this figure can be found at http://osf.io/rsnu4/ and in S2 Data.
(TIF)

**S1 Table. Demographics of AVHs, non-AVHs, and Normal Controls.** The underlying data for this figure can be found at http://osf.io/rsnu4/ and in S3 Data.
(DOCX)

**S1 Data. Data underlying the plots in Fig 2C and 2D.**
(XLSX)

**S2 Data. Data underlying the plots in Figs 3–4 and S1–S5.**
(XLSX)

**S3 Data. Data underlying the plots in Fig 5.**
(XLSX)

**S4 Data. Data underlying the plots in Fig 6.**
(XLSX)

## Acknowledgments

We thank Jiaqiu Sun and Huiling Li for their help in conducting the experiments.

## Author Contributions

**Conceptualization:** Fuyin Yang, Xing Tian.

**Data curation:** Fuyin Yang, Xinyi Cao, Hui Li, Xinyu Fang, Lingfang Yu, Siqi Li, Zenan Wu, Chunbo Li, Chen Zhang.

**Formal analysis:** Fuyin Yang.

**Funding acquisition:** Fuyin Yang, Xing Tian.

**Methodology:** Fuyin Yang, Hao Zhu.

**Supervision:** Fuyin Yang, Chunbo Li, Chen Zhang, Xing Tian.

**Writing – original draft:** Fuyin Yang.

**Writing – review & editing:** Fuyin Yang, Xing Tian.

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
