## [Editor Report · Decision Letter 0]

10 Jul 2024

Dear Dr Tian, 

Thank you for submitting your manuscript entitled "Impaired motor-to-sensory transformation mediates auditory hallucinations" for consideration as a Research Article by PLOS Biology.

Your revised manuscript has now been evaluated by the PLOS Biology editorial staff as well as by an academic editor with relevant expertise and I am writing to let you know that we would like to send your submission back to the original reviewers.

Once your full submission is complete, your paper will undergo a series of checks in preparation for peer review. After your manuscript has passed the checks it will be sent out for review. To provide the metadata for your submission, please Login to Editorial Manager (https://www.editorialmanager.com/pbiology) within two working days, i.e. by Jul 12 2024 11:59PM.

Kind regards,

Christian

Christian Schnell, PhD

Senior Editor

PLOS Biology

cschnell@plos.org

---

## [Decision Letter · Decision Letter 1]

20 Aug 2024

Dear Dr Tian,

Thank you for your patience while we considered your revised manuscript "Impaired motor-to-sensory transformation mediates auditory hallucinations" for publication as a Research Article at PLOS Biology. This revised version of your manuscript has been evaluated by the PLOS Biology editors, the Academic Editor and one of the original reviewers.

Based on the reviews and on our Academic Editor's assessment of your revision, we are likely to accept this manuscript for publication, provided you satisfactorily address the following data and other policy-related requests:

* Please add the links to the funding agencies in the Financial Disclosure statement in the manuscript details.

* DATA POLICY:

Regardless of the method selected, please ensure that you provide the individual numerical values that underlie the summary data displayed in the following figure panels as they are essential for readers to assess your analysis and to reproduce it: 2CD, 3BD, 4BD, 6DEF, S1B, S3BD, S4B and S5BD

* CODE POLICY

* The writing can still be improved at some places. Please carefully check and revise your manuscript for grammar and spelling.

We expect to receive your revised manuscript within two weeks. 

*Published Peer Review History*

*Press*

Sincerely,

Christian

Christian Schnell, PhD

Senior Editor

cschnell@plos.org

PLOS Biology

Reviewer remarks:

Reviewer #1 (Andreea O. Diaconescu): The revisions aimed to address the methodological concerns raised, improve the clarity and specificity of the hypotheses, and place the results of the study within the broader context of existing research.

---

## [Editor Report · Decision Letter 2]

6 Sep 2024

Dear Xing,

Thank you for the submission of your revised Research Article "Impaired motor-to-sensory transformation mediates auditory hallucinations" for publication in PLOS Biology. On behalf of my colleagues and the Academic Editor, Uta Noppeney, I am pleased to say that we can in principle accept your manuscript for publication, provided you address any remaining formatting and reporting issues. These will be detailed in an email you should receive within 2-3 business days from our colleagues in the journal operations team; no action is required from you until then. Please note that we will not be able to formally accept your manuscript and schedule it for publication until you have completed any requested changes.

PRESS

We frequently collaborate with press offices. If your institution or institutions have a press office, please notify them about your upcoming paper at this point, to enable them to help maximize its impact. If the press office is planning to promote your findings, we would be grateful if they could coordinate with biologypress@plos.org. If you have previously opted in to the early version process, we ask that you notify us immediately of any press plans so that we may opt out on your behalf.

Sincerely, 

Christian

Christian Schnell, PhD

Senior Editor

PLOS Biology

cschnell@plos.org